# CausalWorld: A Robotic Manipulation Benchmark for Causal Structure and Transfer Learning

**Ossama Ahmed,**[*1] **Frederik Träuble,**[*2] **Anirudh Goyal,**[3] **Alexander Neitz,**[2]
**Yoshua Bengio,**[3] **Bernhard Schölkopf,**[2] **Stefan Bauer,**[†2] **Manuel Wüthrich**[†2]

[1]ETH Zurich, [2]Max Planck Institute for Intelligent Systems, [3]Mila, University of Montreal

## Abstract

Despite recent successes of reinforcement learning (RL), it remains a challenge for agents to transfer learned skills to related environments. To facilitate research addressing this problem, we propose *CausalWorld*, a benchmark for causal structure and transfer learning in a robotic manipulation environment. The environment is a simulation of an open-source robotic platform, hence offering the possibility of sim-to-real transfer. Tasks consist of constructing 3D shapes from a set of blocks - inspired by how children learn to build complex structures. The key strength of *CausalWorld* is that it provides a combinatorial family of such tasks with common causal structure and underlying factors (including, e.g., robot and object masses, colors, sizes). The user (or the agent) may intervene on all causal variables, which allows for fine-grained control over how similar different tasks (or task distributions) are. One can thus easily define training and evaluation distributions of a desired difficulty level, targeting a specific form of generalization (e.g., only changes in appearance or object mass). Further, this common parametrization facilitates defining curricula by interpolating between an initial and a target task. While users may define their own task distributions, we present eight meaningful distributions as concrete benchmarks, ranging from simple to very challenging, all of which require long-horizon planning as well as precise low-level motor control. Finally, we provide baseline results for a subset of these tasks on distinct training curricula and corresponding evaluation protocols, verifying the feasibility of the tasks in this benchmark.[1]

## 1 Introduction

Benchmarks have played a crucial role in advancing entire research fields, for instance computer vision with the introduction of CIFAR-10 and ImageNet (Krizhevsky et al., 2009; 2012). When it comes to the field of reinforcement learning (RL), similar breakthroughs have been achieved in domains such as game playing (Mnih et al., 2013; Silver et al., 2017), learning motor control for high-dimensional simulated robots (Akkaya et al., 2019), multi-agent settings (Baker et al., 2019; Berner et al., 2019) and for studying transfer in the context of meta-learning (Yu et al., 2019). Nevertheless, trained agents often fail to transfer the knowledge about the learned skills from a training environment to a different but related environment sharing part of the underlying structure. This can be attributed to the fact that it is quite common to evaluate an agent on the training environments themselves, which leads to overfitting on these narrowly defined environments (Whiteson et al., 2011), or that algorithms are com-

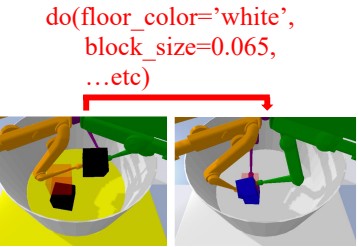

do(floor_color='white', block_size=0.065, …etc)

Figure 1: Example of do-interventions on exposed variables in `CausalWorld`.

---

*Equal contribution.
Correspondence to: <ossama.ahmed@mail.mcgill.ca >, <frederik.traeuble@tuebingen.mpg.de >
†Equal Advising   [1] https://sites.google.com/view/causal-world/home

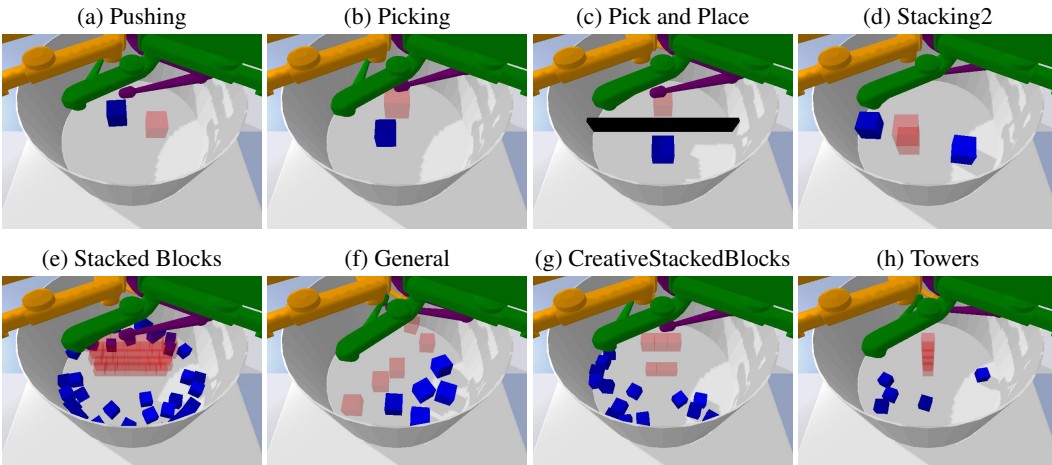

Figure 2: Example tasks from the task generators provided in the benchmark. The goal shape is visualized in opaque red and the blocks in blue.

pared using highly engineered and biased reward functions which may result in learning suboptimal policies with respect to the desired behaviour; this is particularly evident in robotics.

In existing benchmarks (Yu et al., 2019; Goyal et al., 2019a; Cobbe et al., 2018; Bellemare et al., 2013; James et al., 2020) the amount of shared causal structure between the different environments is mostly unknown. For instance, in the Atari Arcade Learning environments, it is unclear how to quantify the underlying similarities between different Atari games and we generally do not know to which degree an agent can be expected to generalize.

To overcome these limitations, we introduce a novel benchmark in a robotic manipulation environment that we call `CausalWorld`. It features a diverse set of environments that, in contrast to previous designs, share a large set of parameters and parts of the causal structure. Being able to intervene on these parameters (individually or collectively) permits the experimenter to evaluate agents' generalization abilities with respect to different types and magnitudes of changes in the environment. These parameters can be varied gradually, which yields a continuum of similar environments. This allows for fine-grained control of training and test distributions and the design of learning curricula.

A remarkable skill that humans learn to master early on in their life is building complex structures using spatial-reasoning and dexterous manipulation abilities (Casey et al., 2008; Caldera et al., 1999; Kamii et al., 2004). Playing with toy blocks constitutes a natural environment for children to develop important visual-spatial skills, helping them 'generalize' in building complex composition designs from presented or imagined goal structures (Verdine et al., 2017; Nath & Szücs, 2014; Dewar, 2018; Richardson et al., 2014). Inspired by this, `CausalWorld` is designed to aid in learning and investigating these skills in a simulated robotic manipulation environment corresponding to the open-source TriFinger robot platform from Wüthrich et al. (2020), which can be built in the real world. Tasks are formulated as building 3D goal shapes using a set of available blocks by manipulating them - as seen in Fig. 1. This yields a diverse familiy of tasks, ranging from relatively simple (e.g. pushing a single object) to extremely hard (e.g. building a complex structure from a large number of objects).

`CausalWorld` improves upon previous benchmarks by exposing a large set of parameters in the causal generative model of the environments, such as weight, shape and appearance of the building blocks and the robot itself. The possibility of intervening on any of these properties at any point in time allows one to set up training curricula or to evaluate an agent's generalization capability with respect to different parameters. Furthermore, in contrast to previous benchmarks (Chevalier-Boisvert et al., 2018; Cobbe et al., 2018), researchers may build their own real-world platform of this simulator at low cost, as detailed in Wüthrich et al. (2020), and transfer their trained policies to the real world.

| Benchmark | do-interventions interface | procedurally generated environments | online distribution of tasks | setup custom curricula | disentangle generalization ability | real-world similarity | open-source robot | low-level motor control | long-term planning | unified success metric |
|---|---|---|---|---|---|---|---|---|---|---|
| RLBench | ✗ | ✗ | ✗ | ✗ | ✗ | ✓ | ✗ | ✓ | ✗ | ✗ |
| MetaWorld | ✗ | ✗ | ✗ | ✗ | ✗ | ✓ | ✗ | ✓ | ✗ | ✗ |
| IKEA | ✗ | ✗ | ✗ | ✗ | ✗ | ✓ | ✗ | ✓ | ✓ | ✓ |
| MuJoBan | ✗ | ✗ | ✗ | ✓ | ✗ | ✓ | ✗ | ✓ | ✓ | ✓ |
| BabyAI | ✗ | ✓ | ✗ | ✗ | ✗ | ✗ | ✗ | ✗ | ✓ | ✓ |
| CoinRun | ✗ | ✓ | ✗ | ✗ | ✗ | ✗ | ✗ | ✗ | ✗ | ✓ |
| AtariArcade | ✗ | ✗ | ✗ | ✗ | ✗ | ✗ | ✗ | ✗ | ✓/✗ | ✓ |
| CausalWorld | ✓ | ✓ | ✓ | ✓ | ✓ | ✓ | ✓ | ✓ | ✓ | ✓ |

Table 1: Comparison of Causal World with RLBench (James et al., 2020), MetaWorld (Yu et al., 2019), IKEA (Lee et al., 2019), BabyAI (Chevalier-Boisvert et al., 2018), CoinRun (Cobbe et al., 2018), AtariArcade (Bellemare et al., 2013), MuJoBan etc. (Mirza et al., 2020),

Finally, by releasing this benchmark we hope to facilitate research in causal structure learning, i.e. learning the causal graph (or certain aspects of it) as we operate in a complex real-world environment whose dynamics follow the laws of physics, which induce causal relations between the variables. Changes to the variables we expose can be considered do-interventions on the underlying structural causal model (SCM). Consequently, we believe that this benchmark offers an exciting opportunity to investigate causality and its connection to RL and robotics.

Our main contributions can be summarized as follows:

- We propose `CausalWorld`, a new benchmark comprising a parametrized family of robotic manipulation environments for advancing out-of-distribution generalization and causal structure learning in RL.

- We provide a systematic way of defining curricula and disentangling generalization abilities of RL agents with respect to different changes in the environment, since we allow for do-interventions to be performed on different environment variables (parameters and states) individually.

- We establish baseline results for some of the available tasks under different learning algorithms, thus verifying the feasibility of the tasks.

- We show how different learning curricula affect generalization across different axes by reporting some of the in-distribution and out-of-distribution generalization capabilities of the trained agents.

## 2 CAUSALWORLD BENCHMARK

Here we make the desiderata outlined in the introduction more precise:

1. The set of environments should be sufficiently diverse to allow for the design of challenging transfer tasks.

2. We need to be able to intervene on different properties (e.g. masses, colors) individually, such that we can investigate different types of generalization.

3. It should be possible to convert any environment to any other environment by gradually changing its properties through interventions; this requirement is important for evaluating different levels of transfer and for defining curricula.

4. The environments should share some causal structure to allow algorithms to transfer the learned causal knowledge from one environment to another.

5. There should be a unified measure of success, such that an objective comparison can be made between different learning algorithms.

6. The benchmark should make it easy for users to define meaningful distributions of environments for training and evaluation. In particular, it should facilitate evaluation of in-distribution and out-of-distribution performance.

7. The simulated benchmark should have a real-world counterpart to allow for sim2real.

In light of these desiderata, we propose a setup in which a robot must build goal shapes using a set of available objects. It is worth noting that similar setups were proposed previously in a less realistic setting, e.g. in (Janner et al., 2018; Bapst et al., 2019; McCarthy et al.; Akkaya et al., 2019; Fahlman, 1974; Winston, 1970; Winograd, 1972). Specifically, a task is formulated as follows: *given a set of available objects, the agent needs to build a specific goal structure*, see Fig. 1 for an example. The vast amount of possible target shapes and environment properties (e.g. mass, shape and appearance of objects and the robot itself) makes this a diverse and challenging setting to evaluate different generalization aspects. `CausalWorld` is a simulated version (using the Bullet physics engine (Coumans et al., 2013)) of the open-source TriFinger robot platform from Wüthrich et al. (2020). Each environment is defined by a set of variables such, as gravity, floor friction, stage color, floor color, joint positions, various block parameters (e.g. size, color, mass, position, orientation), link colors, link masses and the goal shape. See Table 3 in the Appendix for a more extensive list of these variables.

Desideratum 1 is satisfied since different environment properties and goal shapes give rise to very different tasks, ranging from relatively easy (e.g. re-positioning a single cube) to extremely hard (e.g. building a complex structure). Desideratum 2 is satisfied because we allow for arbitrary interventions on these properties, hence users or agents may change parameters individually or jointly. Desideratum 3 is satisfied because the parameters can be changed gradually. Desideratum 4 is satisfied because all the environments share the causal structure of the robot, and one may also use subsets of environments which share even more causal structure. We satisfy desideratum 5 by defining the measure of success for all environments as the volumetric overlap of the goal shape with available objects. Further, by splitting the set of parameters into a set A, intended for training and in-distribution evaluation, and a set B, intended for out-of-distribution evaluation, we satisfy desideratum 6. Finally, since the TriFinger robot (Wüthrich et al., 2020) can be built in the real-world, we satisfy desideratum 7. Desideratum 7 and 2 are in partial conflict since sim2real is only possible for the tasks which are constrained to the variables on which the robot can physically act upon.

**Task generators:** To generate meaningful families of similar goal shapes, `CausalWorld` allows for defining task generators which can generate a variety of different goal shapes in an environment. For instance, one task generator may generate pushing tasks, while another one may generate tower-building tasks (see Fig. 2). Each task generator is initialized with a default goal shape from its corresponding family and comes with a sampler to sample new goal shapes from the same family. Additionally, upon construction, one can specify the environments' initial state and initial goal shape structure when deviating from the default. The maximum episode time to build a given shape is `number_of_blocks × 10` seconds. `CausalWorld` comes with eight pre-defined task generators (see Fig. 2).

- Three generators create goal shapes with a single block: *Pushing* with the goal shape on the floor, *Picking* having the goal shape defined above the floor and *Pick and Place* where a fixed obstacle is placed between the initial block and goal pose.

- *Stacking2* involves a goal shape of two stacked blocks, which can also be considered one instance of the *Towers* generator.

- The remaining generators use a variable number of blocks to generate much more complex and challenging target shapes, as detailed in the appendix: *Towers*, *Stacked Blocks*, *Creative Stacked Blocks* and *General*.

Given that building new environments using current physics simulators is often tedious, we provide a simple API for users who wish to create task generators for new challenging shape families, which may be added to `CausalWorld`'s task generators repository.

**Action and Observation Spaces:** The robot can be chosen to operate in either joint position control mode, joint torque control mode, end-effector position control mode, or the delta of each. In any of these cases, the action is 9-dimensional (one per joint). We provide two observation modes: *structured* and *pixel*. In the *structured* mode, the observation vector is constructed using a rule for the ordering of the relevant variables, such as joint positions, joint velocities, block positions, etc. Thus, the size of the observation space depends on the number of blocks, which could potentially change with every new goal sampled, e.g. in Towers, (Creative) Stacked Blocks and General. In

contrast, in the *pixel* mode, the agent receives six RGB images (hence the dimension of the observation is $6 \times 3 \times 128 \times 128$), the first three images are rendered from the three cameras mounted around the TriFinger robot, and the last three images specify the goal image of the target shape rendered from the same cameras. Additionally, `CausalWorld` allows users to set up a fully customized observation space.

**Rewards:** The reward function $r$ is defined as the fractional volumetric overlap of the blocks with the goal shape, which ranges between 0 (no overlap) and 1 (complete overlap). Since this reward function is shared across all tasks, an agent that learned $r$ from some training tasks could in principle use it to solve unseen goal structures. There is also the possibility of modifying the reward function to 1) sparsify the reward further by returning a binary reward signal instead, or 2) add a dense reward function in order to introduce inductive biases via domain knowledge and solution guidance. We hope that the considerable complexity and diversity of goal shapes motivate and accelerate the development of algorithms that are not dependent on hand-tuned reward functions.

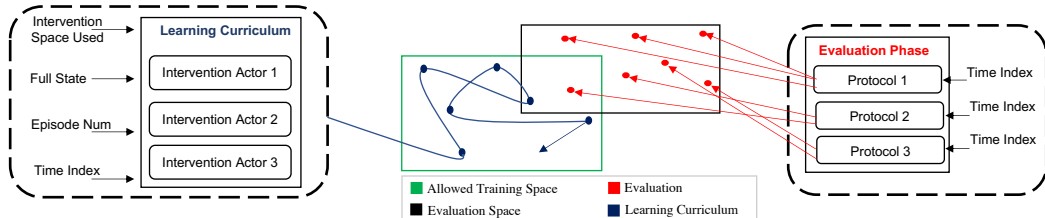

Figure 3: Key components for generic training and evaluation of RL agents. Left: A learning curriculum which is composed of various intervention actors that decide which variables to intervene on (for a valid intervention, values need to be in the allowed training space (ATS)). Right: Evaluation protocols are shown which may intervene on variables at episode resets or within episodes (for a valid intervention, values need to be in the evaluation space (ES)). Middle: we represent the ATS and ES, where each intervention results in one point in the spaces. As shown ATS and ES may intersect, eg. if the protocols are meant to evaluate in-distribution generalization.

**Training and evaluation spaces:** In this benchmark, a learning setting consists of an allowed training space (ATS) and an evaluation space (ES), both of which are subspaces of the full parameter space. During training, in the simplest setting, parameters are sampled iid from the ATS. However, unlike existing benchmarks, `CausalWorld` allows in addition for curricula within the ATS as well as settings where the agent itself intervenes on the parameters within an episode (see Fig. 3). Similarly, during evaluation, parameters may be sampled iid from the evaluation space at each episode reset, or there can be interventions within an episode. Moreover, in order to retrieve the setting considered in most RL benchmarks, we could set the ATS and the ES to be identical and intervene only on object and robot states (and keep other environment properties constant) at each episode reset. However, to evaluate out-of-distribution generalization, one should set the two spaces (ATS and ES) to be different; possibly even disjoint. Additionally, to evaluate robustness with respect to a specific parameter (e.g. object mass), one may define the training and evaluation spaces which only differ in that particular parameter. In order to facilitate the definition of appropriate training and evaluation settings, we pre-define two disjoint sets, $\mathbf{A}_i$ and $\mathbf{B}_i$, for each parameter $i$. Through this, one can for instance define the training space to be $\mathbf{A}_1 \times \mathbf{A}_2 \times ...$ and the evaluation space to be $\mathbf{B}_1 \times \mathbf{B}_2 \times ...$ to assess generalization with respect to all parameters simultaneously. Alternatively, the evaluation space could be defined as $\mathbf{A}_1 \times \mathbf{A}_2 \times ... \times \mathbf{B}_i \times \mathbf{A}_{i+1} \times ...$ to assess generalization with respect to parameter $i$ only. Lastly, users may also define their own spaces which could then be integrated into the benchmark to give rise to new learning settings.

**Intervention actors:** To provide a convenient way of specifying learning curricula, we introduce intervention actors. At each time step, such an actor takes all the exposed variables of the environment as inputs and may intervene on them. To encourage modularity, one may combine multiple actors in a learning curriculum. This actor is defined by the episode number to start intervening, the episode number to stop intervening, the timestep within the episode it should intervene and the episode periodicity of interventions. We provide a set of predefined intervention actors, including

an actor which samples parameters randomly at each episode reset, which corresponds to domain-randomization. It is also easy to define custom intervention actors, we hope that this facilitates investigation into optimal learning curricula (see Fig. 3).

**Probing the Causal Structure in RL:** The problem setting in RL is usually formulated using the language of Markov Decision Processes (MDPs) or Partially Observable Markov Decision Processes (POMDPs) (Sutton & Barto, 1999), but can be also represented by Structural Causal Models (SCMs), as shown in (Buesing et al., 2018), refer to section E in the Appendix for a detailed explanation. This is achieved by formulating all conditional probability distributions as deterministic functions that take independent noise variables as inputs. These independent noise variables can specify different scenarios while the deterministic functions reflect the causal mechanisms of the system (Schölkopf et al., 2021). Changes in the environment can stem from two different sources:

1. The agent may alter the state of the environment (e.g. the position of a block) indirectly, through its actions (e.g. pushing the block by applying appropriate torques at the motors).

2. During the execution of a learning curriculum or an evaluation protocol, we may directly intervene on any variable of the SCM, including all the latent variables of the causal model that are not accessible to the RL agent (such as gravity, object mass or color).

(1) is the default type of admissible interventions in RL benchmarks, whereas CausalWorld allows for interventions of type (2) in addition. The idea is that interventions on these latent variables, e.g. during a learning curriculum, will allow the agent to distinguish between spurious correlations that are only present in a particular setting and true causal relations that will hold across all settings (i.e. interventions). If the agent is able to learn such a representation of the underlying SCM structure, we would expect it to perform well even in out-of-distribution scenarios (Schölkopf et al., 2021; Dittadi et al., 2021) because the causal structure remains the same, even when the functional form of certain relations may vary (e.g. when transferring to the real robot). Moreover, we hope that by having access to a broad range of interventions in CausalWorld it will aid the inference of the underlying SCM structure through the different causal discovery methods (see Figure 4 for a subset of the expected SCM to be learned), which in turn addresses the lack of causal discovery benchmarks for real world challenges.

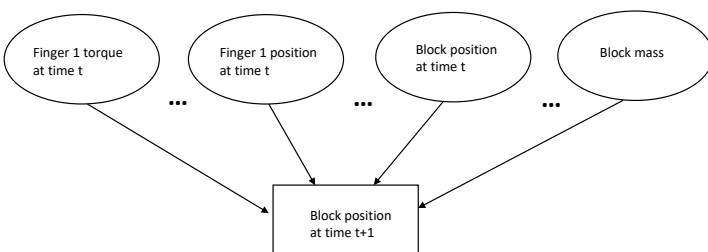

Figure 4: A subset of an SCM represented as a DAG for an environment in CausalWorld with one block on the floor. Here, we only show a subset of the causal variables affecting the block position at time t+1.

## 3 RELATED WORK

Previous benchmarks for RL mostly focused on the single task learning setting such as OpenAI Gym and DM control suite (Tassa et al., 2018; Brockman et al., 2016). In contrast, a recent line of work, e.g. Meta-World and RLBench (Yu et al., 2019; James et al., 2020), aims at studying multi-task learning as well as meta-learning. Such benchmarks mostly provide non-parametric, hand-designed task variations, it is hence unclear how much structure is shared between them. For instance, it is not clear how different it is to "open a door" compared to "opening a drawer". To address the ambiguity in the shared structure between the tasks, `CausalWorld` was designed to allow interventions to be performed on many environment variables giving rise to a large space of tasks with well-defined

relations between them, which we believe is a missing key component to address generalization in RL. A detailed comparison between CausalWorld and similar benchmarks is shown in Table 1.

Similar parametric formulations of different environments were used in the generalization-for-RL literature, which has played an important role in advancing the field (Packer et al., 2018; Rajeswaran et al., 2017; Pinto et al., 2017; Yu et al., 2017; Henderson et al., 2017a; Dulac-Arnold et al., 2020; Chevalier-Boisvert et al., 2018). In these previous works, variables were mostly changed randomly, as opposed to the full control over the variables provided by `CausalWorld`.

Another important open problem for the RL community is the standardization of the reported learning curves and results. RL methods have been shown to be sensitive to a range of different factors (Henderson et al., 2017b). Thus it is crucial to devise a set of metrics that measure reliability of RL algorithms and ensure their reproducibility. Chan et al. (2019) distinguishes between several evaluation modes like "evaluation during training" and "evaluation after learning". Osband et al. (2019) recently proposed a benchmarking suite that disentangles the ability of an algorithm to deal with different types of challenges. Its main components are: enforcing a specific methodology for an agent's evaluation beyond the environment definition and isolating core capabilities with targeted 'unit tests' rather than integrating the general learning ability.

Moreover, causality has been historically studied from the perspective of probabilistic and causal reasoning (Pearl, 2009), cognitive psychology (Griffiths & Tenenbaum, 2005), and more recently in the context of machine learning (Goyal et al., 2019b; Schölkopf et al., 2021; Baradel et al., 2019; Bakhtin et al., 2019). On the contrary, we believe its link to robotics is not yet drawn systematically. To bridge this gap, one of the main motivations of `CausalWorld` was to facilitate research in causal learning for robotics, such as observational discovery of causal effects in physical reality, counterfactual reasoning, and causal structure learning.

## 4    EXPERIMENTS

To illustrate the usage of this benchmark and to verify the feasibility of some basic tasks, we evaluate current state-of-the-art model-free (MF-RL) algorithms on a subset of the goal shape families described in Section 2 and depicted in Fig. 2: (a) Pushing, (b) Picking, (c) Pick and Place, and (d) Stacking2. These goal shapes reflect basic skills that are required to solve more complex construction tasks.

**Setup:**   The idea here is to investigate how well an agent will perform on different evaluation distributions, depending on the curriculum it has been trained with. We train each method under the following curricula:

- Curriculum 0: no environment changes; each episode is initialized from the default task lying in space $\mathbf{A}$ - note that here the initial state never changes (i.e. no interventions).

- Curriculum 1: goal shape randomization; at the beginning of each episode a new goal shape is sampled from space $\mathbf{A}$ (i.e. interventions on goal position and orientation).

- Curriculum 2: full randomization w.r.t. the task variables[5]; every episode a simultaneous intervention on all variables is sampled from space $\mathbf{A}$ (i.e. can be seen as equivalent to extreme domain randomization in one space).

The curriculum will, as expected, affect the generalization capabilities of the trained agents. With `CausalWorld`'s formulation, these generalization capabilities can easily be disentangled and benchmarked quantitatively, as explained in Section 2. For each of the goal shape families (a, b, c, d from Fig. 2), we train agents under the three described curricula using the following MF-RL algorithms: The original Proximal Policy Optimization (PPO) from Schulman et al. (2017), Soft Actor-Critic (SAC) from Haarnoja et al. (2018) and the Twin Delayed DDPG (TD3) from Fujimoto et al. (2018). We provided these methods with a hand-designed dense reward function as we did not observe any success with the sparse reward only. Each of the mentioned setups is trained for five different random seeds, resulting in 180 trained agents.

---

[1]  [5] Note that each task generator can suppress interventions that would yield goal shapes outside its family.

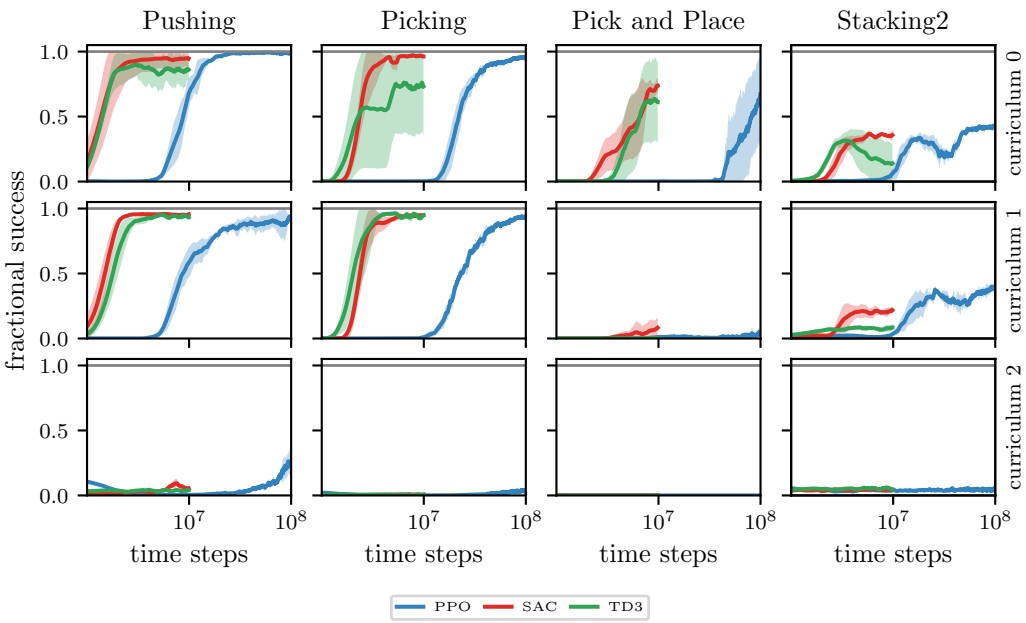

Figure 5: Fractional success curves averaged over five random seeds for the tasks and learning algorithms specified above, under three different training curricula: (0) no curriculum, (1) goal position and orientation randomization in space **A** every episode and (2) a curriculum where we intervene on all variables in space **A** simultaneously every episode.

**Training model-free RL methods:**    We report the training curves averaged over the random seeds in Fig. 5. As can be seen from these fractional success training curves, MF-RL methods are capable of solving the single block goal shapes (pushing, picking, pick and place) seen during training time given enough experience. However, we observe that none of the methods studied here managed to solve stacking two blocks. The score below 0.5 indicates that it only learns to push the lower cube into the goal shape. This shows that multi-object target shapes can become nontrivial quickly and that there is a need for methods making use of the modular structure of object-based environments. To no surprise, the training curriculum has a major effect on learning. For example, methods rarely manage to pick up any significant success signal under extreme domain randomization as in curriculum 2, even after 100 million timesteps. Note that these curves represent the scores under the conditions of the training environments. Next, we will discuss shared evaluation protocols that allow to benchmark and compare agents trained under different conditions.

**Benchmarking generalization capabilities along various axes:**    For each of the four goal shape families, we define a set of 12 evaluation protocols that we consider meaningful and representative for benchmarking the different algorithms. In the protocols presented here, we sample the values from a protocol-specific set of variables at the start of each episode while keeping all other variables fixed to their default values. We evaluate each agent on 200 episodes by computing the fractional success score at the last time step of each episode and reporting the mean. These evaluation protocols allow to disentangle generalization abilities, as they show robustness with respect to different types of interventions, see Fig. 6. The following are some of the observations we made for pushing:

- Agents that were trained on the default pushing task environment (curriculum 0) do well (as expected) on the default task (P0). Interestingly, we likewise see a generalization capability to initial poses from variable space A (P4). This can be explained by a substantial exploration of the block positions via manipulation during training. In contrast, we see that the agents exhibit weaknesses regarding goal poses (P5), they seem to overfit to their training settings in this case.

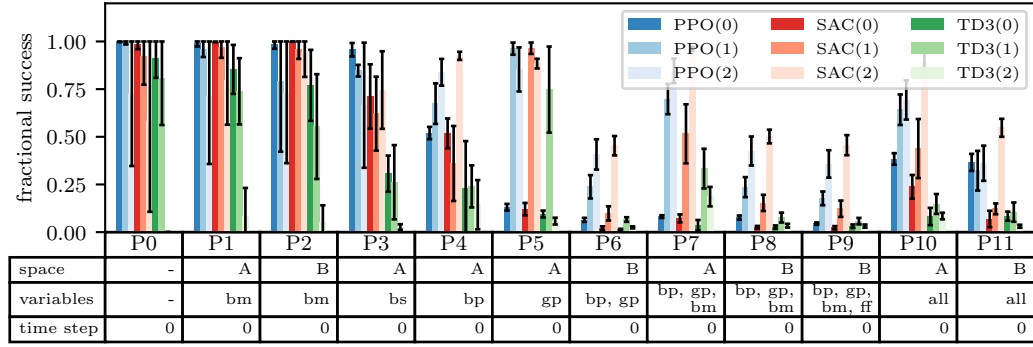

Figure 6: Evaluation scores for pushing baselines. Each protocol was evaluated for 200 episodes and each bar is averaged over five models with different random seeds. The variables listed under each protocol are sampled from the specified space at the start of every episode while all other variables remain fixed [bp block pose, bm block mass, bs block size, gp goal pose, ff floor friction].

- For agents trained with goal pose randomization (curriculum 1) we see similar results as with curriculum 0, with the difference that agents under this curriculum generalize robustly to different goal poses (P5), as one would expect.
- Finally, agents that experience extreme domain randomization (curriculum 2) at training time, fail to learn any relevant skill as shown by the flat training curve in Fig. 5. An explanation for this behavior could be that the agent might need more data and optimization steps to handle this much more challenging setting. Another possibility is that it may simply not be possible to find a strategy which simultaneously works for all parameters (note that the agent does not have access to the randomized parameters and hence must be robust to them). This poses an interesting question for future work.

As expected, we observe that an agent's generalization capabilities are related to the experience gathered under its training curriculum. `CausalWorld` allows us to explore this relationship in a differentiated manner, assessing which curricula lead to which generalization abilities. This will not only help uncover an agent's shortcomings but may also aid in investigating novel learning curricula and approaches for robustness in RL. Lastly, we note that this benchmark comprises extremely challenging tasks that appear to be out of reach of current model free methods without any additional inductive bias.

## 5 CONCLUSION

We have introduced a new benchmark - `CausalWorld` - to facilitate research in causal structure and transfer learning using a simulated environment of an open-source robot, where learned skills could potentially be transferred to the real world. We showed how allowing for interventions on the environment's properties yields a diverse familiy of tasks with a natural way of defining learning curricula and evaluation protocols that can disentangle different generalization capabilities. A natural extension of our work is to develop new RL algorithms that focuses on out of distribution generalization (whether its a different subspace of the state space or a completely different task).We hope that the flexibility and modularity of `CausalWorld` will allow researchers to easily define appropriate benchmarks of increasing difficulty as the field progresses, thereby coordinating research efforts towards ever new goals.

## 6 ACKNOWLEDGMENTS

The authors would like to thank Felix Widmaier, Vaibhav Agrawal and Shruti Joshi for the useful discussions and for the development of the TriFinger robot's simulator (Joshi et al., 2020), which served as a starting point for the work presented in this paper. AG is also grateful to Alex Lamb and Rosemary Nan Ke for useful discussions. We thank the International Max Planck Research School for Intelligent Systems (IMPRS-IS) for supporting FT.

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

# 7 APPENDIX

## A OBSERVATIONS

Observations in `CausalWorld` has two modes, "structured" and "pixel". When using "pixel" mode, 6 images are returned consisting of the current images rendered from 3 different views on top of the TriFinger platform, showing the current state of the environment, as well as the 3 equivalent goal images rendered from the same points of view, showing the goal shape that the robot have to build by the end of the episode.

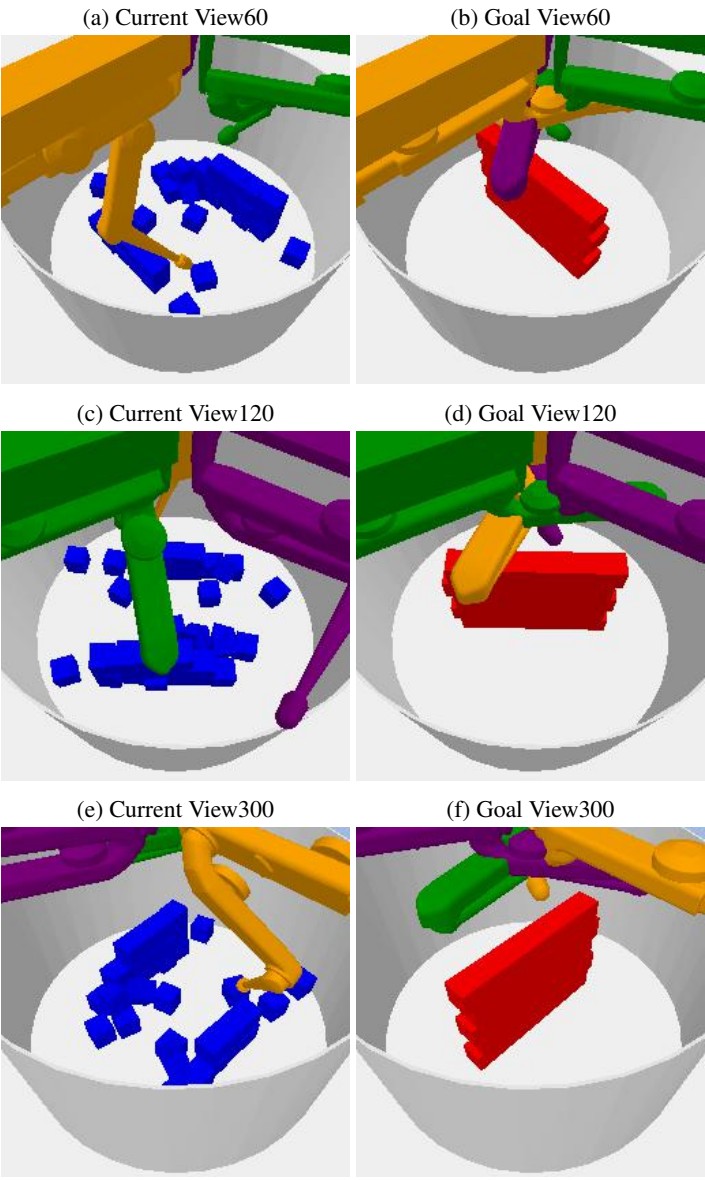

Figure 7: Example "pixel" mode observations returned at each step of the environment.

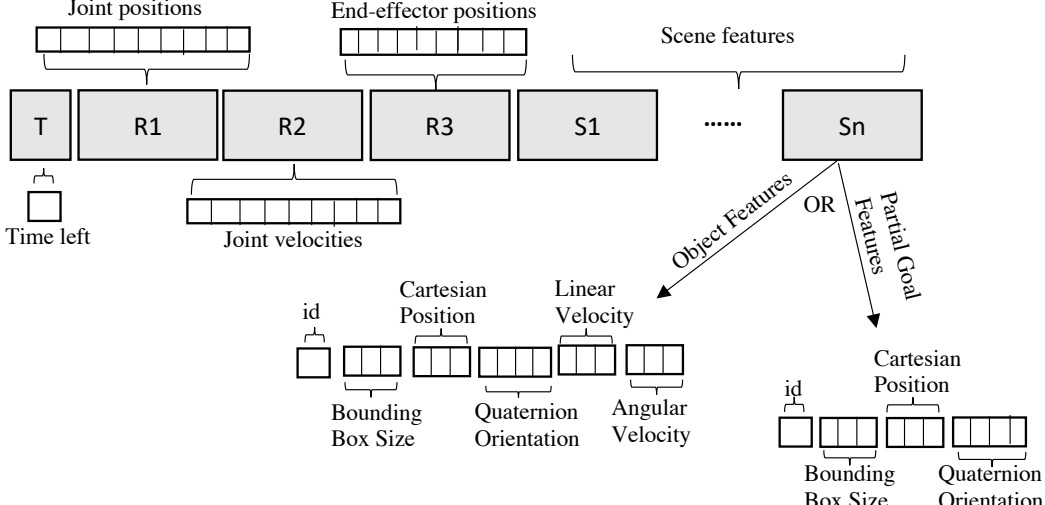

Figure 8: Structured observation description. For the scene features, all the blocks feature vector are concatenated first. Following that the partial goals feature vector are concatenated in the same order. Lastly, if there is any obstacles/ fixed blocks, their feature vectors are concatenated at the end following the same description as the partial goal features.

## B    TRIFINGER PLATFORM

The robot from (Wüthrich et al., 2020) shown in figure 9 is open-sourced and can be reproduced and built in any research lab; since its inexpensive (about $5000), speeding up sim2real research.

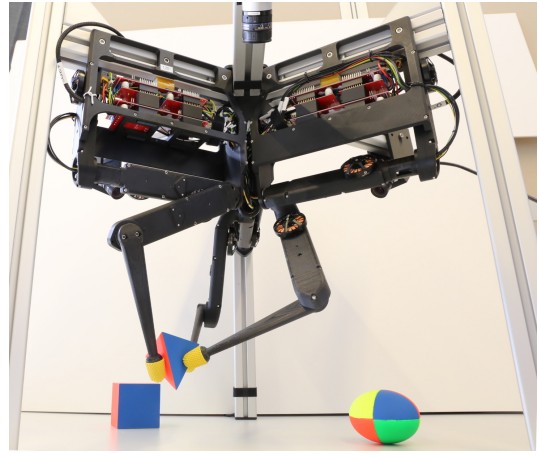

Figure 9: The TriFinger platform.

## C    TASK GENERATORS

1. *Pushing*: task where the goal is to push one block towards a goal position with a specific orientation; restricted to goals on the floor level.

2. *Picking*: task where the goal is to pick one block towards a goal height above the center of the arena; restricted to goals above the floor level.

3. *Pick And Place*: task where the arena is divided by a fixed long block and the goal is to pick one block from one side of the arena to a goal position with a variable orientation on the other side of the fixed block.

4. *Stacking2*: task where the goal is to stack two blocks above each other in a specific goal position and orientation.

5. *Towers*: task where the goal is to stack multiple n blocks above each other in a specific goal position and orientation - exactly above each other creating a tower of blocks.

6. *Stacked Blocks*: task where the goal is to stack multiple n blocks above each other in an arbitrary way to create a stable structure. The blocks don't have to be exactly above each other; making it more challenging than the ordinary towers task since the its harder to come up with a stable structure that covers the goal shape volume.

7. *Creative Stacked Blocks*: exactly the same as the *Stacked Blocks* task except that the first and last levels of the goal are the only levels shown or "imposed" and the rest of the structure is not explicitly specified, leaving the rest of the goal shape to the imagination of the agent itself; this is considered the most challenging since its it needs the agent to understand how to build stable structures and imagine what can be filled in the middle to connect the two levels in a stable way.

8. *General*: the goal shape is an arbitrary shape created by initially dropping an arbitrary number of blocks from above the ground and waiting till all blocks come to a rest position where this becomes the goal shape that the agent needs to fill up afterwards.

| Variable | Sub Variable | Space **A** | Space **B** |
|---|---|---|---|
| gravity[z] | - | $[-10, -7]$ | $[-7, -4]$ |
| floor friction | - | $[0.3, 0.6]$ | $[0.6, 0.8]$ |
| stage friction | - | $[0.3, 0.6]$ | $[0.6, 0.8]$ |
| stage color [rgb] | - | $[0.0, 0.5]^3$ | $[0.5, 1]^3$ |
| floor color [rgb] | - | $[0.0, 0.5]^3$ | $[0.5, 1]^3$ |
| joint positions | - | $[[-1.57, -1.2, -3.0]^3, [-0.69, 0, 0]^3]$ | $[[-0.69, 0, 0]^3, [1.0, 1.57, 3.0]^3]$ |
| block | size | $[0.055, 0.075]^3$ | $[0.075, 0.095]^3$ |
| block | color | $[0.0, 0.5]^3$ | $[0.5, 1]^3$ |
| block | mass | $[0.015, 0.045]$ | $[0.045, 0.1]$ |
| block | position (cylindrical) | $[[0, -\pi, h/2], [0.11, \pi, 0.15]]$ | $[[0.11, -\pi, h/2], [0.15, \pi, 0.3]]$ |
| goal cuboid | size | $[0.055, 0.075]^3$ | $[0.075, 0.095]^3$ |
| goal cuboid | color | $[0.0, 0.5]^3$ | $[0.5, 1]^3$ |
| link | color | $[0.0, 0.5]^3$ | $[0.5, 1]^3$ |
| link | mass | $[0.015, 0.045]$ | $[0.045, 0.1]$ |

Table 2: Description of a subset of the high level variables, exposed in CausalWorld, and their corresponding spaces, $h$ refers to the height of the block.

| Task Generator | Variable | Space **A** | Space **B** |
|---|---|---|---|
| Picking | goal height | $[0.08, 0.20]$ | $[0.20, 0.25]$ |
| Towers | tower dims | $[[0.08, 0.08, 0.08], [0.12, 0.12, 0.12]]$ | $[[0.12, 0.12, 0.12], [0.20, 0.20, 0.20]]$ |

Table 3: Example of task generators' specific high level variables, exposed in CausalWorld, and their corresponding spaces. For a full list of each task generators' variables and their corresponding spaces, please refer to the documentation at (https://sites.google.com/view/causal-world/home).

| Task generators | Dense reward |
|---|---|
| Pushing | $-750\Delta^t(o_1, e) - 250\Delta^t(o_1, g_1)$ |
| Picking | $-750\Delta^t(o_1, e) - 250\Delta^t(o_{1,z}, g_{1,z}) - 125\Delta^t(o_{1,x,y}, g_{1,x,y}) - 0.005\|v^t - v^{t-1}\|$ |
| Pick and Place | $-750\Delta^t(o_1, e) - 50\Delta^t(o_{1,x,y}, g_{1,x,y}) - 250(|o_{1,z}^t - t| - |o_{1,z}^{t-1} - t|) - 0.005\|v^t - v^{t-1}\|$ |
| Stacking | $\mathbf{1}_{d^t(o_1,e)>0.02}(-750\Delta^t(o_1, e) \quad - \quad 250\Delta^t(o_1, g_1)) \quad +$ $\mathbf{1}_{d^t(o_1,e)<0.02}(-750\Delta^t(o_2, e) - 250(|o_{2,z}^t - g_{2,z}^t| - |o_{1,z}^{t-1} - g_{2,z}^t|) -$ $\mathbf{1}_{o_{2,z}^t - g_{2,z}^t>0} 125\Delta^t(o_{2,x,y}, g_{2,x,y})) - 0.005\|v^t - v^{t-1}\|$ |

Table 4: Description of the dense rewards applied in our experiments. The following notation was applied: $v^t \in \mathbf{R}^3$ joint velocities, $e_i^t \in \mathbf{R}^3$ i-th end-effector positions, $o_i^t \in \mathbf{R}^3$ i-th block position, $g_i^t \in \mathbf{R}^3$ i-th goal block position, $d^t(o, e) = \sum_i \|e_i^t - o^t\|$ the distance between end-effectors and the block, $\Delta_{o,e}^t = d^t(o, e) - d^{t-1}(o, e)$ the distance difference w.r.t. the previous timestep. The target height parameter $t$ for pick and place is 0.15 if block and goal are of different height. Otherwise, $t$ is half the goal height.

# D    TRAINING DETAILS

The experiments were carried out using the `stable baselines` implementation of PPO, SAC and TD3. We used a 2 layer MLP Policy [256,256] for all the policies. PPO was trained on 20 workers up to 100 million timesteps in parallel and SAC as well as TD3 were trained serially for 10 million timesteps.

| PPO | | SAC | | TD3 | |
|---|---|---|---|---|---|
| discount | 0.99 | discount | 0.95 | discount | 0.96 |
| batch size | 120000 | entropy coeff | 1e-3 | batch size | 128 |
| learning rate | 2.5e-4 | batch size | 256 | learning rate | 1e-4 |
| entropy coef. | 0.01 | learning rate | 1e-4 | buffer size | 500000 |
| value function coef. | 0.5 | target entropy | auto | tau | 0.02 |
| gradient clipping (max) | 0.5 | buffer size | 1000000 | | |
| n minibatches per update | 40 | tau | 0.001 | | |
| n training epochs | 4 | | | | |

Table 5: Learning algorithms hyper parameters used in the baselines experiments.

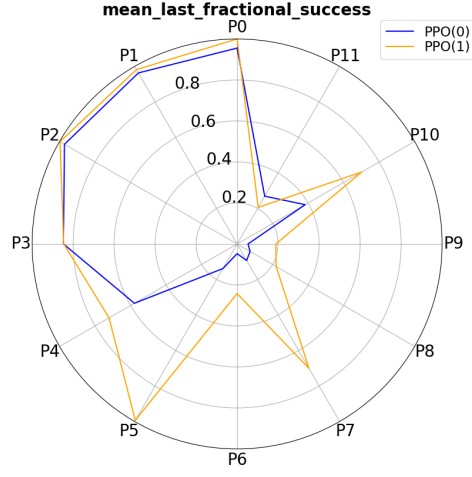

Figure 10: An example of model selection in `CausalWorld` by evaluating generalization across the various axes using the previously mentioned protocols. Here we compare two agents trained on different curricula using PPO.

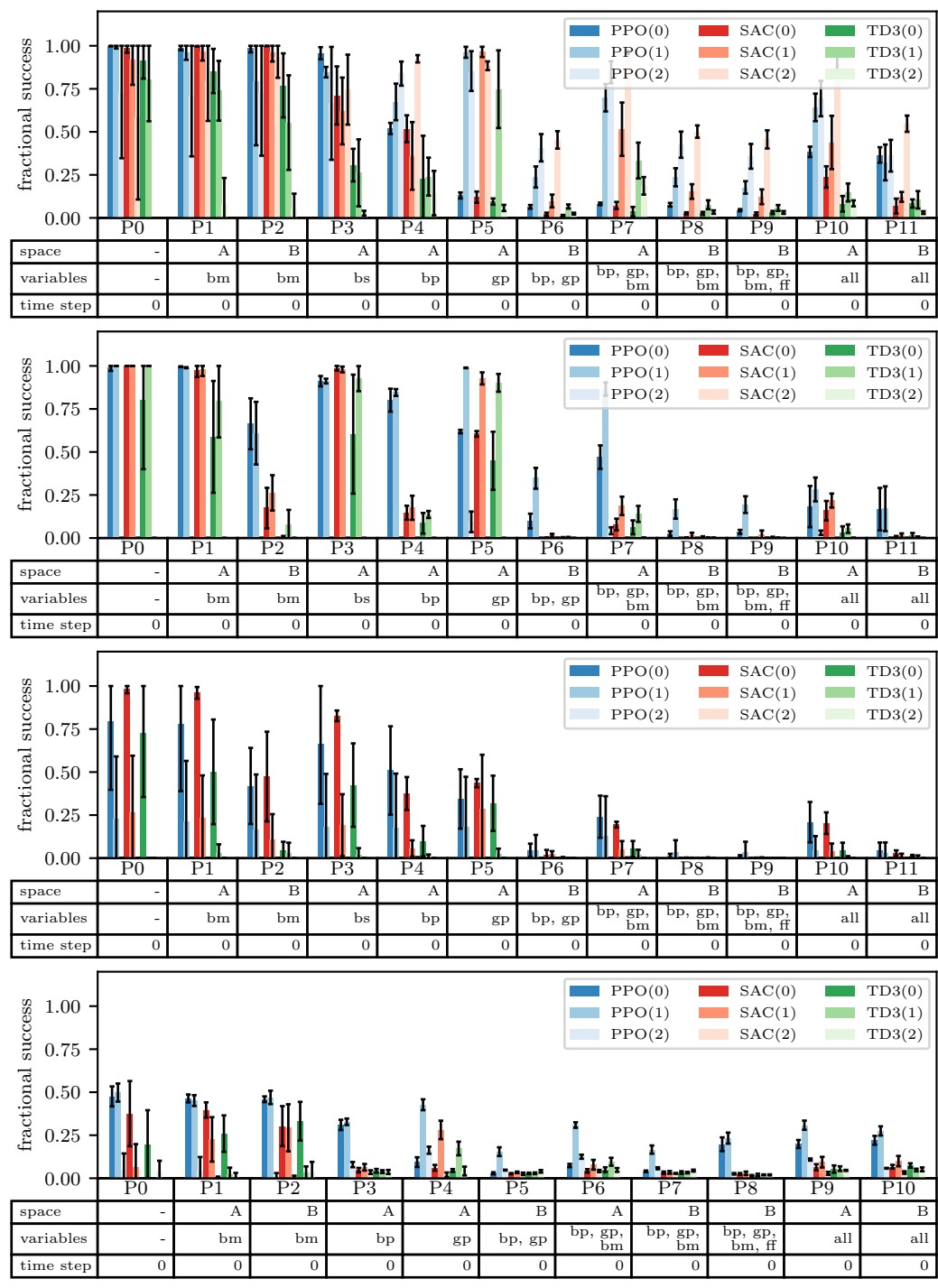

Figure 11: Evaluation scores, for pushing, picking, pick and place and stacking2 baselines, from top to bottom respectively. Each protocol was evaluated for 200 episodes and each bar is averaged over five models with different random seeds [bp block pose, bm block mass, bs block size, gp goal pose, ff floor friction].

# E    CAUSALITY IN REINFORCEMENT LEARNING

**Definition 1:**    A partially observable Markov decision processes (POMDP) is defined by the tuple $(S, A, T, R, \Omega, O, \gamma, \rho_0, H)$ with states $s \in S$, actions $a \in A$ and observations $o \in \Omega$ determined by the state and action of the environment $O(o|s, a)$. $T(s_{t+1}|s_t, a_t)$ is the transition probability distribution function, $R(s_t, a_t)$ is the reward function, $\gamma$ is the discount factor, $\rho_0(s)$ is the initial state distribution at the beginning of each episode, and $H$ is the time horizon per episode. The objective of RL algorithms is to learn a policy $\pi(a_t|h_t)$ with history $H_t = (O_1, A_1, ..., A_{t-1}, O_t)$ that maximizes the discounted expected reward $J(\pi) = \mathbb{E}^{\pi}\left[\sum_{t=0}^{H}\gamma^t r_t\right]$.

**Definition 2:**    A structural causal model (SCM) $M$ over $X = (X_1, ..., X_N)$ is given by a DAG $G$ over nodes $X$, independent noise RVs $U = (U_1, ..., U_N)$ with distributions $P_{U_i}$ and functions $f_1, ..., f_N$ such that $X_i = f_i(pa_i, U_i)$, where $pa_i \subset X$ are the parents of $X_i$ in $G$. An SCM entails a distribution $P$ with density $p$ over $(X, U)$.

**Definition 3:**    An intervention $I$ in an SCM $M$ consists of replacing the RHS $f_i(pa_i, U_i)$ by $f_i^I(pa_i^I, U_i)$ with $i \subseteq \{1, ..., N\}$ where $pa_i^I$ are the parents in a new DAG $G^I$. The resulting SCM is denoted with $M^{do(I)}$ with distribution $P^{do(I)}$ and density $p^{do(I)}$.

**Representing POMDPs as SCMs:**    A POMDP can be presented as an SCM $M$ by unrolling the POMDP through time by expressing all conditional distributions, e.g the transition kernel $P_{S_{t+1}|S_t, A_t}$, as deterministic functions (which reflects the causal mechanisms of the system) with independent noise variables $U$, such as $S_{t+1} = f_{st}(S_t, A_t, U_{st})$. This is always possible using auto-regressive formalization as shown in (Buesing et al., 2018). The distribution $P^{\pi}$ over trajectories $T$ determined by the SCM, which means running a different policy $\mu$ (instead of $\pi$) in the environment is an intervention by itself $I(\pi \to \mu)$ resulting in model distribution $P^{do(I(\pi\to\mu))}$. Given the SCM, one can reason about the alternative outcomes of different actions or even different environment properties (such as friction) (known as counterfactual inference) which is not possible if we only model the system as a conditional distribution $P_{O|A, U_c}$.

**Generalizing agents:**    An agent should generalize well to changes in the environment if it achieves an equal performance after an intervention $I$ is performed on the SCM $M$ resulting in $M^{do(I)}$. This can be accomplished for instance as explained in (Schölkopf et al., 2021), by reusing the learned causal mechanisms. For example, if an intervention was performed on the colour of one of the available blocks in the environment, it won't change the underlying causal mechanisms. Thus, this sort of interventions on a non causal variable shouldn't affect the resulting actions of the agent. Nevertheless, this is often not the case with RL agents trained using visual observations. (Alver & Precup, 2020) showed recently that algorithms which were designed specifically for Meta-RL can display strong overfitting when they are evaluated on challenging visual tasks. On the other hand, if an intervention was performed on a causal variable with respect to a policy $\pi$, such as the mass of an object, the resulting action sequence will be expected to change for robust control. Additionally, if an intervention is performed on some of the goal variables, the reward function might also change and a robust controller should also be able to react to it accordingly.

