# OpenReview forum: "CausalWorld: A Robotic Manipulation Benchmark for Causal Structure and Transfer Learning"
_ICLR.cc/2021/Conference — ICLR 2021 Poster_

### Official Review · AnonReviewer2 · 2020-10-27
**Useful benchmark for studying generalization of RL**

**Rating:** 6
**Confidence:** 3

**Review:**

This paper proposed a new benchmark for studying reinforcement learning and its generalization in the context of the robotic manipulation problem. To study the generalization of a learned policy, the proposed benchmark is equipped with an interface that makes intervention easy. This interface helps to define a training space and an evaluation space so that one can systemically study both in-distribution and out-of-distribution generalization of a learned policy. At the same time, the proposed benchmark simulates an open-source robot platform, which makes sim2real transfer experiments easier.

Strengths:
* This paper proposed an RL benchmark with many good properties: systematic intervention of environment distribution and potential application to sim2real transfer experiments
* The source code of the benchmark provided with the submission is clean and well documented, so it could benefit future research based on this work.
* Proposed evaluation protocol gives an insight on how this benchmark can be used to evaluate generalization of RL agent.

Weaknesses:
* One concern I have about this benchmark is that the training difficulty may hinder the analysis of generalization. It is expected that a rich training distribution would lead to better generalization, but it seems rich training distribution makes training difficult and results in worse performance on evaluation distribution.
* As far as I understand, there are a few predefined tasks, and each task distribution cannot be intervened by modifying some task-relevant parameters. However, I can imagine parameterizing the tasks. For example, we can parameterize push task distribution by introducing a range of (x, y) position of block initial positions, and introducing a range of distance from initial position to goal position. If I have misunderstood the detail of the task parameterization, please elaborate on this point.


Questions to authors:
* I had the impression that this paper attempts to make a connection with the research about causality. For example, the name of the benchmark is "CausalWorld", and the text mentions "opportunity to investigate causality", and "underlying structural causal model (SCM)". However, it is unclear to me how this benchmark can help to study causality exactly. Could you elaborate on this point?
* It seems that space A and space B in Table 2 (Appendix C) is an arbitrary split of a range. How these ranges are determined? Is there any motivation behind this split?


Recommendation:
I recommend accepting this paper because it could benefit the field by helping people to easily study generalization in a complex robotic manipulation setting. To the best of my knowledge, no existing open-sourced RL environment for robotic manipulation does not support systematic intervention of the environment distribution.

---

> ### Author Response · Authors · 2020-11-16
> **Reviewer2 Response - Thanks for your feedback**
>
> We thank the reviewer for the feedback which will allow us to improve the paper!
>
> “One concern I have about this benchmark is that the training difficulty may hinder the analysis of generalization. It is expected that a rich training distribution would lead to better generalization, but it seems rich training distribution makes training difficult and results in worse performance on evaluation distribution”
>
> → Our intention with the experiments is to showcase the flexibility regarding curricula and performance evaluation schemes offered with CausalWorld, rather than solving new tasks or proposing new algorithms. The space of possible environments which can be created using CausalWorld is very large. Hence there are a myriad of interesting experiments and curricula one could set up, such that we can only give some examples:
>
> The training curricula are chosen to show the two extremes of (0) a completely static environment and (2) all environment variables being randomized after each episode reset (which is indeed expected to be a hard and difficult setting to learn skills in) and some intermediate curriculum (1). In the same way, one can easily define other curriculums that can represent less difficult training distributions to begin with. In Fig 5 we show how CausalWorld can be used to evaluate performance of agents in a differentiated manner: Rather than just computing a single score, we can assess generalization ability with respect to changes in individual (or groups of) parameters and how it relates to different training curricula.
>
> “As far as I understand, there are a few predefined tasks, and each task distribution cannot be intervened by modifying some task-relevant parameters. However, I can imagine parameterizing the tasks. For example, we can parameterize push task distribution by introducing a range of (x, y) position of block initial positions, and introducing a range of distance from initial position to goal position. If I have misunderstood the detail of the task parameterization, please elaborate on this point.”
>
> → We fully agree that intervening on the task-relevant parameters and parameterizing each task distribution with a range of values are indeed helpful in such a benchmark and actually that's exactly what CausalWorld offers.
>
> A full description is provided on page 5 under “Training and Evaluation Spaces” and a subset of the parameterizations regarding each task distribution as well as the interventions allowed are defined in table 2 in the appendix.
> E.g. for the mass of a block across all tasks, space A is  [0.015, 0.045]  and space B is [0.045, 0.1]; the split was chosen to make sure that the tasks are feasible in both spaces. A subset of the full list is defined in table 2 in the appendix.
>
> If we take pushing as an example (see Fig 5): Agents are trained with interventions on the goal pose in space A (curriculum 1) and end up interpolating to different goal poses coming from the same distribution as shown in the results for P5 (as expected).
> Confirming the reviewers’ intuition, the example given regarding (x,y) position  would be a valid use case of the benchmark.
>
> “I had the impression that this paper attempts to make a connection with the research about causality. For example, the name of the benchmark is "CausalWorld", and the text mentions "opportunity to investigate causality", and "underlying structural causal model (SCM)". However, it is unclear to me how this benchmark can help to study causality exactly. Could you elaborate on this point?”
>
> →  As we provide the tools to perform interventions on the causal and non-causal variables in the environment, many of the current causal structure learning algorithms from the literature that require intervention capabilities could be evaluated in settings that build upon CausalWorld. Also, by assessing generalization with respect to changes in causal variables (e.g. mass), we indirectly assess whether the agent has learned a causal notion of said variables (e.g. mass).
> We will make sure to elaborate on this point further in the manuscript.
>
> “It seems that space A and space B in Table 2 (Appendix C) is an arbitrary split of a range. How these ranges are determined? Is there any motivation behind this split?”
>
> → The splits are chosen such that the task is solvable in each of the two spaces (A and B); so that reasonable conclusions about transferability and generalization could be made from the experiments. Naturally, there are many other valid ways to split the parameter space, in fact users have the option to define their own custom intervention spaces.

---

> > ### Author Response · Authors · 2020-11-19
> > **Anything else you'd like us to respond to?**
> >
> > Once again, we thank the reviewer for their feedback and valuable comments that will help us improve this paper.
> >
> > Since the first phase of response period is over, if you have time and could indicate if there are any other concerns of yours which we have not addressed, we'd be happy to take a look.
> >
> > Thanks for your time.

---

> > > ### Comment · AnonReviewer2 · 2020-11-21
> > > **I'm confused about the experiment too.**
> > >
> > > Thanks to the authors for the detailed comments about the questions raised in the review.
> > >
> > > Now I have a better understanding of the environment and I still believe that this flexible environment would benefit the field.
> > > However, I agree with other reviewers that it is hard to understand why Figure 4 and Figure 5 matter and how they help to understand the main motivation behind this new environment design.
> > >
> > > What confuses me in Figure 4 is that "full randomization" should generalize better, but the result only shows that "full generalization" doesn't learn. With this result, it is hard to disentangle issues of "curriculum" for learning training distribution well, and issues of "training distribution" for generalizing to testing distribution well.
> > >
> > > I think what Figure 4 suggests might be that someone may design a train / test distribution split to test generalization for a particular aspect of the environment, but this attempt may completely fail just because they don't have a good curriculum to train on the training distribution. My comment “One concern I have about this benchmark is that the training difficulty ...." was pointing out the difficulty of disentangling "learning difficulty" and "generalization difficulty", and the feedback didn't fully resolve my concern.
> > >
> > > I'm curious whether it is possible to disentangle "learning difficulty" and "generalization difficulty" and separately measuring them. When someone studies "curriculum", it is to resolve "learning difficultY". If someone tried to study "generalization difficulty", at least we should make sure that learning on training distribution should be doable.

---

> > > > ### Author Response · Authors · 2020-11-23
> > > > **Experiments Clarification**
> > > >
> > > > Once again, we thank the reviewer for their feedback and valuable comments that will help us improve this paper.
> > > >
> > > > “What confuses me in Figure 4 is that "full randomization" should generalize better, but the result only shows that "full generalization" doesn't learn. With this result, it is hard to disentangle issues of "curriculum" for learning training distribution well, and issues of "training distribution" for generalizing to testing distribution well.”
> > > >
> > > > → We argue that by explicitly defining different spaces for each of the exposed variables and subsequently designing evaluation protocols that test agent’s performance across both spaces, we take a step towards disentangling issues related to the curriculum’s in-distribution generalization and out-of-distribution generalization.
> > > >
> > > >  We agree with the reviewer that when training with curriculum 2 (extreme random randomization), the policy is expected to generalize better. However, the primary insight about the generalization on test distribution w.r.t train distribution (curriculum) can be drawn from the first two curriculums with their respective evaluations. The results from the extreme full randomization curriculum (curriculum 2 ), however, are obscured by the unknown confounder: how much compute one would need to solve them and if this is possible for the MLP policies used in this work; given that not all randomized variables are observed.
> > > >
> > > > Additionally, our results highlight that it is not straightforward to study why curriculum 2 is not generalizing as expected without solving other important challenges in RL, e.g. sample efficiency.
> > > >
> > > > “I think what Figure 4 suggests might be that someone may design a train / test distribution split to test generalization for a particular aspect of the environment, but this attempt may completely fail just because they don't have a good curriculum to train on the training distribution.”
> > > >
> > > > → We fully agree that the explanation made by the reviewer is correct. This is why we believe that CausalWorld offers a broad set of tools by customizing evaluation protocols to investigate these failure modes. We hoped that the evaluation protocols we reported present these many different insights. Future experiments and study can define custom protocols that might be better suited for their setting.
> > > >
> > > > “My comment “One concern I have about this benchmark is that the training difficulty ...." was pointing out the difficulty of disentangling "learning difficulty" and "generalization difficulty", and the feedback didn't fully resolve my concern.”
> > > >
> > > > → Indeed it's difficult to disentangle the “learning difficulty” and the “generalization difficulty” completely since they are both tightly coupled. What we attempt here is to measure both independently such that investigations towards optimal curricula that generalize well could be made.
> > > >
> > > > "I’m curious whether it is possible to disentangle "learning difficulty" and "generalization difficulty" and separately measure them. When someone studies "curriculum", it is to resolve "learning difficulty". If someone tried to study "generalization difficulty", at least we should make sure that learning on training distribution should be doable"
> > > >
> > > > →  We believe that an agent’s performance on a testing distribution will always be coupled to the training distribution. We welcome any suggestions to achieve such disentanglement and would be happy to include it in the benchmark. The benchmark currently offers full flexibility in defining different curricula and evaluation protocols.
> > > >
> > > > The experiments are chosen here to primarily showcase the flexibility regarding curricula and performance evaluation schemes offered with CausalWorld.
> > > >
> > > > If there are any other concerns of yours which we have not addressed, we'd be happy to clarify them further.

---

### Official Review · AnonReviewer3 · 2020-10-28
**Official Blind Review #3**

**Rating:** 4
**Confidence:** 4

**Review:**

### Summary

Motivated by the difficulty of evaluating RL’s ability to transfer behaviors across environments, the authors propose the CausalWorld benchmark. Unlike prior benchmarks, CausalWorld exposes well-defined casual variables, in the form of task factors, and focuses on robotic manipulation of an open-source robot platform. The authors make CausalWorld easily usable for both training (in defining a learning curriculum) and evaluation (in targeting specific expected generalizations), and make it easy to extend. In their original release, the authors include eight concrete tasks to test generalization, and present baseline results on these tasks.

---

### Positives

- The paper is strongly motivated and tackles a real and practical problem. Evaluating transfer in RL agents has been challenging, especially for robotics, and the authors’ proposed benchmark framework could be useful in addressing this.
- The authors’ benchmark supports useful behavior both for training, in gradually varying the task distributions, and in testing, for evaluating generalization ability. Additionally, since it is tied to a real world, open source robot platform, this benchmark in the future could also be used to evaluate sim2real transfer.
- The authors’ framework is defined in a way that seems easy-to-extend and supports multiple use cases, including custom “task generators” for defining new tasks or goals, and “intervention actors” to define a learning curriculum.
The paper provides relatively strong baseline experiments, with quantitative results across several model-free RL algorithms (PPO, SAC, TD3), and multiple potential curriculum techniques.

---

### Negatives

The current iteration of the experiments, with figures 4 and 5, are unclear and confusing. Specifically:

- In figure 4, it’s unclear to me what the new experimental result is here, given that the benchmark is meant to test transfer and generalization ability, and the results presented are on training curves. It seems that the main conclusion here is that the choice of learning curriculum is important for performance, which as the authors point out is unsurprising?
- In figure 4, curriculum 2 seems poorly motivated, in that full randomization without any curriculum at the beginning of training seems likely to fail as it does. Have the authors considered testing curriculums where the domains for the causal variables get progressively more challenging? For instance, automatic domain randomization (ADR) from “Solving rubik’s cube with a robot hand” may be a useful curriculum to compare.
- It was challenging for me to follow Figure 5, since it was not clear what the training environments agents were being trained on, and which environments they were being evaluated under. Further, do these results for pushing hold across other tasks (picking, pick and place, stacking2)?
- In figure 5, what is the time step reported (0 across all evaluations)?
- Figure 5 shows that there is some generalization to tasks in space A and B, but it is unclear how A and B differ, what the variables between both are, which environments in A were trained on, and what P0-P11 are.

In addition I think some other experimental results would be helpful:

- The authors could investigate a more detailed analysis on different reward structures, other progressive curriculums, and other methods that claim better generalization and transfer performance.
- Some qualitative results exploring the relationship between the identified and proposed causal variables (potentially through the lens of the agent performance) would be helpful.
- A qualitative experiment comparing the difference in learned behaviors between two policies and the difference in performance, to show that the performance reported by the benchmark does match intuition would be helpful.

Apart from the experimental results, I have some other broader (and potentially less pressing) concerns:

- The authors limit their framework and results to the manipulation of simple block shapes. Manipulating non-block objects would result in more complex goals, introducing more causal variables that are potentially harder to disentangle and represent cleanly, but are still important for real-world applications.
- The authors are motivated by facilitating research in causal structure learning, but this paper focuses almost exclusively on studying transfer learning and generalization ability. Potentially this is mitigated by benchmarking causal learning algorithms that try to directly learn the causal graph or reason between causal variables. For instance, potentially the causal graph-parameterized policy learning approach from “Causal Confusion in Imitation Learning”, or similar algorithms, would be good to include.

---

### Recommendation

Overall, I vote for rejecting. I think the benchmark is well motivated, but not backed up with strong experimental results. The motivation for the benchmark is to show that the framework can be used to study transfer performance, but the current experimental results do not convince me that the framework makes it easy to uncover new insights in practice. One reason to potentially accept the benchmark is that it seems easy to extend, but this is also difficult to evaluate from the limited experiments presented.

If the authors were to respond to some of my comments above, by providing a better understanding of the figures and experiments (in case I am misinterpreting the current results), and by showing the utility of the benchmark, then some of my concerns would be addressed.

---

### Minor feedback

- Which hand designed dense reward function is being used? I see this is present in the supplementary figures, I would also add a reference in the main text.
- Which observation spaces were the figures trained/evaluated in (state or pixel)?

----

---

> ### Author Response · Authors · 2020-11-14
> **Reviwer3 Response Part 1 - Thanks for your feedback**
>
> We thank the reviewer for the feedback which will allow us to improve the paper!
> An important takeaway from your review is that we need to be clearer and more explicit about the motivation of the experiments. Would adding something along the lines of the following paragraph be helpful?
>
> Our intention with these experiments is to showcase the flexibility regarding curricula and performance evaluation schemes offered with CausalWorld, rather than solving new tasks or proposing new algorithms. The space of possible environments which can be created using CausalWorld is very large. Hence there are a myriad of interesting experiments and curricula one could set up, here we can only give some examples:
> The training curricula are chosen to show the two extremes of (0) a completely static environment and (2) all environment variables being randomized after each episode reset and some intermediate curriculum (1). In Fig 5 we show how CausalWorld can be used to evaluate performance of agents in a differentiated manner: Rather than just computing a single score, we can assess generalization ability with respect to changes in individual (or groups of) parameters and how it relates to different training curricula.
>
> “In figure 4, it’s unclear to me what the new experimental result is here, given that the benchmark is meant to test transfer and generalization ability, and the results presented are on training curves. It seems that the main conclusion here is that the choice of learning curriculum is important for performance, which as the authors point out is unsurprising?”
>
> → While ultimately we care about transfer and generalization ability (figure 5), we believe it is still important to also show training curves. This allows to see e.g. whether the agents picked up any success signal and whether they converged, which might explain part of the evaluation performance in figure 5. We will make sure to clarify this point further in the paper. Additionally, since we are presenting a novel benchmark, providing training results can be a useful reference for other researchers for reproducibility.
>
>  “In figure 4, curriculum 2 seems poorly motivated, in that full randomization without any curriculum at the beginning of training seems likely to fail as it does. Have the authors considered testing curriculums ..like ADR..”
>
> → We agree with the reviewer that curriculum 2 was expected to fail, however, the primary motivation behind the choice of the curriculums is to showcase the generalization capacity of two extreme cases of interventions on the environment variables and a standard engineered curriculum as mentioned before.
> Indeed ADR would be an interesting direction to explore in a follow-up work - thanks for pointing it out. The curriculums chosen are provided as baselines to prove the feasibility of some tasks in the benchmark, rather than engineering a robust policy.
>
> ““It was challenging for me to follow Figure 5, since it was not clear what the training environments agents were being trained on, and which environments they were being evaluated under.”
>
> → Thanks for pointing out the potential confusion in figure 5; we updated the figure and caption accordingly (please look at figure 5 and figure 6 in the current pdf). In our setting, we don’t explicitly define training and testing environments but rather we define two distributions (A and B) for each of the exposed variables in the environment. So the training and testing environments are defined by the curriculum, the evaluation protocols and the corresponding spaces for interventions. During training, space A is enabled for interventions (but that doesn’t mean the agent will be trained on all the values in A).
> E.g. in pushing, consider the mass of the block: space A is [0.015, 0.045] and space B is [0.045, 0.1], so potentially depending on the chosen curriculum the agent can experience many values in A by having many interventions on the environment accordingly. In curriculum 0 and 1 the agent only explores one value for the mass (0.02), since there are no interventions on the mass. On the other hand, curriculum 2 explores many random values from space A interval due to the random interventions in this space.
>
> During evaluation, the current evaluation protocols test the agent against space A and space B (depends on the protocol), where some of the values were potentially seen during training. For instance protocol P0 tests the default setting of the task with no interventions performed on the environment (so this one for sure is seen during training). Another example P2 defines an evaluation protocol for random interventions sampled from space B of the block mass - which was not seen during training for the curriculums discussed.
>
> Similar results for the evaluation of picking, pick and place and stacking2 are shown in the appendix(Figure 11).
> We hope the figure and caption modification clarified this point more.

---

> > ### Author Response · Authors · 2020-11-14
> > **Reviwer3 Response Part 2**
> >
> > “In figure 5, what is the time step reported (0 across all evaluations)?”
> >
> > → Thanks for pointing out that this might be a potential confusion. We changed the paper accordingly. The timestep 0 specified in figure 5 is the timestep at which the interventions of each protocol are performed. However, the timestep for the score calculation is the last timestep.
> >
> > “It is unclear how A and B differ..and what P0-P11 are”
> >
> > →  A description of these spaces is provided in page 5 under "Training and evaluation spaces".
> > E.g. for the mass of a block, space A is  [0.015, 0.045] and space B is [0.045, 0.1]; this was chosen arbitrarily to make sure that the tasks are feasible in both spaces. A subset of the full list is defined in table 2 in the appendix.
> > So for pushing for instance as shown in figure 5: agents trained with interventions on the goal pose in space A (curriculum 1) end up interpolating to different goal poses coming from the same distribution as shown in the results for P5 (as expected).
> > P0-P11 are the evaluation protocols used: They are defined by interventions performed on specific variables to evaluate the agent. We can make the analogy here to an examiner who is testing the agent across different axes by intervening on the different environment variables in a specific way. The current protocols P0-P11 uses random interventions on specific variables using one of the spaces A or B.
> >
> > “The authors could investigate .. different reward structures, other progressive curriculums..”
> >
> > → The primary goal of this work is to provide the tools to build and evaluate generalizable agents in a more systematic fashion, rather than building generalizable agents for the tasks specified. The reward structures and the curriculums used are meant to serve as baselines for future comparison to other methods. Therefore, we leave exploring different methods that could potentially have better generalization for future work, since this was not the focus of this work.
> >
> > “Some qualitative results exploring the relationship between the identified and proposed causal variables (potentially through the lens of the agent performance) would be helpful”
> >
> > →The RL agents evaluated in this work did not identify causal variables explicitly, hence an explicit comparison of agents’ internal representations with the actual causal variables would be difficult. Nevertheless, by assessing generalization with respect to changes in causal variables (e.g mass), we indirectly assess whether the agent has learned a notion of the causality of said variables (e.g. mass) with respect to task success.
> >
> > “A qualitative experiment comparing the difference in learned behaviors between two policies and the difference in performance, to show that the performance reported by the benchmark does match intuition would be helpful”
> >
> > → Provided in the supplementary website https://sites.google.com/view/causalworld-iclr/home, under “disentangling generalization” two demonstrations are shown for two policies, one trained with no interventions (curriculum 0) and the other is trained with interventions on goal pose (curriculum 1). In the middle, a radial plot showing the different protocol scores of the two policies overlayed on top of each other. As can be seen, the policy trained with curriculum 0, ended up overfitting on the goal pose and the other one was able to generalize better to different goal poses (expected). With the tools we provide in CausalWorld, this could be measured more explicitly through measuring generalization across different axes.
> >
> > “The authors limit their framework and results to the manipulation of simple block shapes.”
> >
> > → We fully agree with the reviewer, what we present here is the first, but already extensive, iteration of the framework. In follow-up iterations, more causal variables will be exposed for instance. The framework was also designed with a focus on modularity and extensibility, to allow users to define their own block shapes, intervention spaces, task distributions, etc. In the attached code, there are already several tutorials showing how to accomplish some of these extensions. Additionally, some of the tasks provided are already very challenging to solve so it might not be necessary to consider more complex cases.
> >
> > “The authors are motivated by facilitating research in causal structure learning.. potentially the causal graph-parameterized policy learning approach from “Causal Confusion in Imitation Learning”or similar algorithms, would be good to include”
> >
> > → We fully agree that benchmarking with methods that directly learn the causal graph might be very insightful. However, as mentioned before, the focus here is to provide the tools to build and evaluate agents that could generalize. We leave exploring these specific methods for future work.
> > Thanks for pointing out “Causal Confusion in Imitation Learning” as a potential method. However, this work uses imitation learning rather than learning a policy from scratch without prior data.

---

> > > ### Author Response · Authors · 2020-11-19
> > > **Anything else you'd like us to respond to?**
> > >
> > > Once again, we thank the reviewer for their feedback and valuable comments that will help us improve this paper.
> > >
> > > Since the first phase of response period is over, if you have time and could indicate if there are any other concerns of yours which we have not addressed, we'd be happy to take a look.
> > >
> > > Thanks for your time.

---

### Official Review · AnonReviewer5 · 2020-11-05

**Rating:** 8
**Confidence:** 4

**Review:**

This paper presents a new benchmark, CausalWorld, for studying generalization, transfer learning, and causal structure learning in RL and robotics. This is a hugely important problem, and I think this benchmark has some clear advantages over existing benchmarks. The benchmark consists of a simulated three finger robot over a bin containing blocks, within which there are 8 "families" of tasks, (a) pushing, (b) picking, (c) pick and place, (d) stacking 2 blocks, (e) stacking many blocks, (f) general rearrangement, (g) more complex multi-block stacking, and (h) building towers. More importantly, for each family of tasks, there is controllable procedural generation of goals as well as controllable factors of the environment such as object sizes, masses, frictions, colors, etc.

This enables what I think is the key contribution of this paper - a procedural way to define training/evaluation splits where each split samples from different subspaces of the above controllable factors. This provides a systematic way of defining problems which require varying degrees of generalization, measuring the difficulty of such splits, and defining curricula within each split, which is critical to developing learning algorithms which are capable of this sort of generalization. While prior work (Yu et al, James et al) have defined many robotic tasks with some shared structure, one challenge is that it is difficult to say how much generalization one can expect between any two tasks which can be quite different, a problem which this benchmark takes a step towards addressing.

Like the paper mentions, prior works have also used procedural generation over similar controllable factors like this paper does. In fact most physics simulators do allow varying these parameters directly. But, a simple API with a standardized interface to define these splits, as well as common splits that are used as benchmarks is still missing, and this paper takes an important step towards that. While I don't see a link to the code, I would encourage the authors to design their API in a simple and standardized way, as that is likely what would motivate people to use CausalWorld instead of manually defining train/eval splits in their own physics simulators.

The main weakness I see of the paper is the experimental section. The authors train a few SOTA RL algorithms on 3 different train configurations of increasing randomization, and test on 12 different eval configurations. First in terms of clarity, I found Figure 5 difficult to interpret. I think it would be helpful if for each of the Eval protocols it was clearly described what was changing. In general I think the best way to present this would be to look at each pair of "train domain, eval domain" and the corresponding performance, with some clear description about what sort of generalization is needed. Also in terms of performance, it seems like when faced with anything more challenging than push/pick-place with limited randomization, all the SOTA algorithms fail even on the train domain (and as a result struggle on Eval domains as well). So one concern is that of the many domains presented in the benchmark, perhaps only a few are actually solvable by current RL algorithms during training. At the same time I think this indicates the challenges in learning generalizable policies, and may inspire better RL algorithms.

Overall, I think this is an exciting benchmark, and would be excited to use it.

---

> ### Author Response · Authors · 2020-11-16
> **Reviwer5 Response - Thanks for your feedback**
>
> We want to thank the reviewer for their time spent and useful comments that will help us improve this paper. We appreciate your comments and that you find the benchmark exciting and would be willing to use it.
>
> “While I don't see a link to the code, I would encourage the authors to design their API in a simple and standardized way, as that is likely what would motivate people to use CausalWorld instead of manually defining train/eval splits in their own physics simulators.”
>
> → We thank the reviewer for pointing this out. The code base can be accessed under the following link https://drive.google.com/file/d/19wNBbwQkJyZBnbPOWvg6ZNGCRCwi5glj/view. The framework was also designed with a focus on simplicity, modularity and extensibility, to allow users to define their own block shapes, intervention spaces, task distributions, etc.
>
> An example of intervening on the environment:
>
>         task = generate_task(task_generator_id='stacked_blocks')
>
>         env = CausalWorld(task=task, enable_visualization=True)
>
>         env.reset()
>
>         for _ in range(10):
>
>               for i in range(200):
>
>                      obs, reward, done, info = env.step(env.action_space.sample())
>
>               goal_intervention_dict = env.sample_new_goal()
>
>              success_signal, obs = env.do_intervention(goal_intervention_dict)
>
>         env.close()
>
>
> “The main weakness I see of the paper is the experimental section. The authors train a few SOTA RL algorithms on 3 different train configurations of increasing randomization, and test on 12 different eval configurations. First in terms of clarity, I found Figure 5 difficult to interpret. I think it would be helpful if for each of the Eval protocols it was clearly described what was changing.”
>
> → Thanks for pointing out the potential confusion in figure 5; we updated the figure and caption accordingly (please look at figure 5 and figure 6 in the current pdf).
>
> “In general I think the best way to present this would be to look at each pair of "train domain, eval domain" and the corresponding performance, with some clear description about what sort of generalization is needed.”
>
> → We thank the reviewer for this suggestion. Indeed, presenting each pair as “train domain and eval domain” would be very helpful. However, we have 4 task distributions with 3 train domains for each and 12 test domains for each pair. Therefore,  we did not want to discuss 144 train-eval pairs separately but decided to draw conclusions only on some particular interesting examples. Nevertheless, we agree that the protocols and curriculums required some additional clear descriptions which we now added to our manuscript.
>
> “Also in terms of performance, it seems like when faced with anything more challenging than push/pick-place with limited randomization, all the SOTA algorithms fail even on the train domain (and as a result struggle on Eval domains as well). So one concern is that of the many domains presented in the benchmark, perhaps only a few are actually solvable by current RL algorithms during training. At the same time I think this indicates the challenges in learning generalizable policies, and may inspire better RL algorithms”
>
> → One of our key motivations behind this work was to point out limitations in some of the most commonly used  SOTA RL algorithms by proposing environment domains that can be extremely challenging. Although we hypothesize that some of the more challenging  tasks might be still solvable for them, given enough reward engineering and computation resources, we are happy to see that the reviewer agrees with us that it may indicate the challenges in learning generalizable policies and may inspire better RL algorithms.

---

### Official Review · AnonReviewer4 · 2020-11-09
**An interesting benchmark for causal structure and transfer learning based on simulation of a manipulation environment.**

**Rating:** 7
**Confidence:** 3

**Review:**

This paper proposes a a robotic manipulation benchmark for causal structure and transfer learning in a simulation environment considering 3D shape construction tasks given a set of blocks. Baseline results using model-free algorithms are provided for chosen tasks, e.g. pushing, picking, pick&place, stacking. It is also stated that a real version of the robot can be built (as it is open-sourced) for sim2real research. The paper is clearly written, nicely structured and, presents interesting and important ideas. It exposes a large set of parameters, e.g. properties of blocks (size, mass, pose), friction, goals for generalisation evaluations. Having a real-world counterpart makes it very valuable for sim2real research. Authors provide and discuss the relevant previous work detailing how their work connects to the existing literature.  A minor comment: The particular choice of the robot can be motivated, as it  is a special design.

---

> ### Author Response · Authors · 2020-11-16
> **Reviwer4 Response  -  Thanks for your feedback**
>
> We want to thank the reviewer for their time spent and useful comments that will help us improve this paper. We appreciate your comment that you find the benchmark clearly written, valuable for sim-to-real research and nicely structured with interesting and important ideas.
>
> Some additional details on the motivation of the robot: We use the TriFinger robot from Wüthrich et al (2020) https://arxiv.org/abs/2008.03596 where setup and choice of the design is specified extensively. We decided for this setup as it is specifically designed to allow for dexterous fine manipulation beyond grasping, and because it is open-source (which will allow researchers to build their own instance and investigate sim-to real). Also, learning such control as opposed to the much simpler setting with a robotic gripper allows for much more sophisticated skills and capabilities in solving the proposed tasks and hopefully even more challenging ones in the future.
>
> We will make sure to add this to our manuscript.

---

### Author Response · Authors · 2020-11-16
**General Clarification**

We want to thank all the reviewers for their time spent and useful comments that will help us improve this paper!

A clarification on our choice of experiments: our intention with the experiments is to showcase the flexibility regarding curricula and performance evaluation schemes offered with CausalWorld, rather than solving new tasks or proposing new algorithms.

The space of possible environments which can be created using CausalWorld is very large. Hence there are a myriad of interesting experiments and curricula one could set up, here we can only give some examples: The training curricula are chosen to show the two extremes of (0) a completely static environment and (2) all environment variables being randomized after each episode reset and some intermediate curriculum (1). In Fig 5 we show how CausalWorld can be used to evaluate performance of agents in a differentiated manner: Rather than just computing a single score, we can assess generalization ability with respect to changes in individual (or groups of) parameters and how it relates to different training curricula.

---

### Decision · Program_Chairs · 2021-01-07
**Final Decision**

**Decision:**

Accept (Poster)

**Comment:**

CausalWorld is a benchmark for robotic manipulation to address transfer and structural learning. The benchmark includes (i) a variety of tasks (picking, pushing, tower, etc) relating to manipulating blocks, (ii) configurable properties for environments (properties of blocks, gravity, etc), (iii) customizable learning settings involving intervention actors, which can change the environment to induce a curriculum.

The reviewers found the paper compelling and with many strengths, including ‘interesting and important ideas’ (R4), ‘simple API with a standardized interface’ for ‘procedural generation of goals’ (R5), ‘strongly motivated and tackles a real and practical problem’ (R3), and ‘benchmark with many good properties’ (R2). By and large, the reviewers agree that the paper presents an important benchmark satisfying several desiderata, which I certainly agree with.

On the other hand, most of the reviewers (3 out of 4) also raised serious concerns, more prominently, about the experimental results and the causal inference component. For instance, R5 commented that “all the SOTA algorithms fail,” and it is hard to quantify how agents would perform well in different tasks. R3 pointed out the lack of “qualitative results exploring the relationship between the identified and proposed causal variables,” emphasizing that ‘the benchmark is well-motivated, but not backed up with strong experimental results.‘’ R2 identified the lack of clear causal component in the paper while the paper mentions “opportunity to investigate causality” and “underlying structural causal model (SCM).” All in all, these are valid concerns.

The authors' rebuttal was quite detailed, and appreciated, but left some important questions unanswered.  The first and critical issue is about the causal nature of the simulator. The simulator's name is "causalworld" and its stated goal is to provide "a benchmark for causal structure and transfer learning in a robotic manipulation environment."  Also, the first bullet in the list of contributions is: "We propose CausalWorld, a new benchmark comprising a parametrized family of robotic manipulation environments for advancing out-of-distribution generalization and causal structure learning in RL." After reading the paper, I was quite surprised to realize there is no *single* example of a causal model, in any shape or form (e.g., SCM, DAG, Physics) or a structural learning benchmark. In other words, there is a serious, somewhat nontrivial gap between the claimed contributions and what was realized in the paper. One way to address this issue would be to make the causality more explicit in the paper, for example, by sharing the underlying structural causal model, how variables form causal relationships, what causal structures are being learned, and how these learned structures compare with the ground truth. I think these would be reasonable expectations of a simulator that aims to disentangle the causal aspect of the learning process.

The second issue is about the experimental results in terms of generalizability. The authors emphasized on different occasions that "The primary goal of this work is to provide the tools to build and evaluate generalizable agents in a more systematic fashion, rather than building generalizable agents for the tasks specified," or "the experiments is to showcase the flexibility regarding curricula and performance evaluation schemes offered with CausalWorld, rather than solving new tasks or proposing new algorithms." These responses are somewhat not satisfactory given that the goal of the paper is providing tools to build generalizable agents, while the authors seem to suggest they are not committed to actually building such agents. Specifically, the experiments did not demonstrate the simulator as a benchmark but only showcased its flexibility (i.e., offering a large number of degrees of freedom). One suggestion would be to evaluate how algorithms (agents) with varying degrees of "generalizability" power perform across tasks with various difficulty levels. As it currently stands, the tasks are too easy or too hard for the standard, uncategorized algorithms, which makes it difficult to learn any lessons from running something in the simulator.

Lastly, I should mention that the work has a great potential to introduce causal concepts and causal reasoning to robotics, there is a natural and compelling educational component here. Still, the complete absence of *any* discussion of causality and the current literature results hurt this connection and the realization of this noble goal. I believe that after reading the paper, the regular causal inference researcher will not be able to understand what assumptions and types of challenges are entailed by this paper and robotics research. On the other hand, the robotics researcher will not be able to understand what a causal model is and the tools currently available in causal reasoning that may be able to help solve the practical challenges of robotics. In other words, this is a huge missed opportunity since there is a complementary nature of what the paper is trying to do in robotics and the results available in causal inference. I believe readers expect and would benefit from having this connection clearly articulated and realized in a more explicit fashion.

If the issues listed above are addressed, I believe the paper can be a game-changer in understanding and investigating robotics & causality.  Given the aforementioned potential and reasons, I recommend the paper's acceptance *under the assumption that* the authors will take the constructive feedback provided in this meta-review into account and revise the manuscript accordingly.